# Optimizing Traffic Flow in Smart Cities: Soft GRU-Based Recurrent Neural Networks for Enhanced Congestion Prediction Using Deep Learning

Sura Mahmood Abdullah [1], Muthusamy Periyasamy [2], Nafees Ahmed Kamaludeen [3], S. K. Towfek [4,5], Raja Marappan [6], Sekar Kidambi Raju [6,*], Amal H. Alharbi [7] and Doaa Sami Khafaga [7]

1 Department of Computer Sciences, University of Technology, Baghdad 110066, Iraq
2 Department of Cyber Security, Paavai Engineering College (Autonomous), Namakkal 637018, India
3 Department of Computer Science, Jamal Mohamed College (Autonomous), Bharathidasan University, Tiruchirappalli 620020, India
4 Department of Communications and Electronics, Delta Higher Institute of Engineering and Technology, Mansoura 35111, Egypt
5 Computer Science and Intelligent Systems Research Center, Blacksburg, VA 24060, USA
6 School of Computing, SASTRA Deemed University, Thanjavur 613401, India
7 Department of Computer Sciences, College of Computer and Information Sciences, Princess Nourah bint Abdulrahman University, P.O. Box 84428, Riyadh 11671, Saudi Arabia
* Correspondence: sekar_kr@cse.sastra.ac.in

**Abstract:** Recently, different techniques have been applied to detect, predict, and reduce traffic congestion to improve the quality of transportation system services. Deep learning (DL) is becoming increasingly valuable for solving critiques. DL applications in transportation have been collected in several recently published surveys over the last few years. The existing research has discussed the cloud environment, which does not provide timely traffic forecasts, which is the cause of frequent traffic accidents. Thus, a solid understanding of the difficulties in predicting congestion is required because the transportation system varies widely between non-congested and congested states. This research develops a bi-directional recurrent neural network (BRNN) using Gated Recurrent Units (GRUs) to extract and classify traffic into congested and non-congested. This research uses a bidirectional recurrent neural network to simulate and forecast traffic congestion in smart cities (BRNN). Urban regions worldwide struggle with traffic congestion, and conventional traffic control techniques have failed miserably. This research suggests a data-driven approach employing BRNN for traffic management in smart cities, which uses real-time data from sensors and linked devices to control traffic more efficiently. The primary measures include predicting traffic metrics such as speed, weather, current, and accident probability. Congestion prediction performance has also been improved by extracting more features such as traffic, road, and weather conditions. The proposed model achieved better measures than the existing state-of-the-art methods. This research also explores an overview and analysis of several early initiatives that have shown promising results; moreover, it explores two potential future research approaches to increase the accuracy and efficiency of large-scale motion prediction.

**Keywords:** congestion prediction; traffic congestion; transportation systems; recurrent neural networks; bidirectional neural; traffic load; deep learning; gated recurrent unit




## 1. Introduction

Traffic congestion is now a major issue in cities worldwide due to the fast rise of urbanization. Traffic congestion can result in financial losses and significantly lower local citizens' quality of life. To solve this issue, the idea of "smart cities" has been put out, which optimizes the performance of many systems inside a city, including transportation, using cutting-edge technology such as the Internet of Things (IoT), artificial intelligence (AI), and

machine learning (ML). Accurately forecasting traffic congestion is one of the main issues in smart cities when optimizing traffic flow. Accurate traffic jam predictions make routing or changing traffic signals to prevent or lessen congestion feasible.

Recurrent neural networks (RNNs), which can represent sequential data, have gained popularity recently due to DL techniques' promise in traffic prediction applications. In this work, a GRU-based smooth RNN model is developed to enhance congestion prediction in smart cities. A soft gating mechanism is incorporated into the soft GRU (Gated Recurrent Unit), which is a variation of the conventional GRU that improves the handling of missing or noisy input. The suggested model combines historical traffic data with information from other sources such as weather and events to forecast future congestion levels. The performance of the proposed model is examined using an actual traffic dataset and compared to several reference models. The experimental findings support the suggested concept.

Ensuring economic growth and the comfort of road users are two prerequisites for the country's development, which is not possible without traffic flow [1–3]. With the development of the transport sector through traffic information authorities pay more attention to traffic volume monitoring. Traffic forecasts give authorities time to plan resource allocations to ensure a smooth journey. The usefulness of street systems is limited by congestion. The reductions result in both direct and indirect costs for the community [4–6]. The effects of congestion on the economic system and social structure have been extensively studied. Late working hours are a direct result of traffic jams. It was later calculated that the United States lost 8.8 billion work hours to congestion and traffic in any given year. The traffic prediction problem can be defined as estimating parameters related to traffic levels, for example, from 15 min to several hours, using various AI methods using the collected traffic data. Five parameters are typically evaluated when monitoring and forecasting congestion: traffic volume, traffic volume, occupancy, congestion rate, and travel time. Depending on the type of data collected, different AI approaches are used to evaluate the overload parameters [7–9].

A significant study field, particularly in AI and ML, has resulted from the ability to predict traffic congestion in recent times. Over the past few decades, this research field has dramatically expanded due to the emergence of massive data from static sensors or probed navigation systems [10,11]. Several traffic factors are evaluated to anticipate traffic congestion, particularly short-term congestion problems. The majority of studies on anticipating traffic congestion use past data. A few papers, however, have predicted congestion problems in real time. Congestion forecasting is a more complicated problem to solve from the perspective of modifiability than traffic flow prediction in non-congested circumstances. Traffic controllers can implement relief measures thanks to an alert system. Over the years, the infrastructure for collecting traffic data has improved. Researchers studying transportation may now use DNN predictions for this field, thanks to this advancement and the expansion of computational power [12,13].

Previous techniques use empirical or ML algorithms to anticipate incoming traffic to use these advantages. They use algorithms based on features gathered and real-time traffic data as to reveal and preserve fundamental traffic conditions using human-crafted characteristics. Nevertheless, in actuality, various circumstances, such as traffic laws, environmental conditions, and so forth, can affect incoming traffic. It has been established that these individually chosen parameters fall short of fully describing traffic data, making it impossible to make an accurate prediction. DL theory has advanced significantly thanks to the extraordinary amount of data and the speed with which these data can be processed. DL has received a lot of attention because of its extraordinary capacity to dynamically extract characteristics from massive amounts of source data. It has already been effectively used in several disciplines, including object and speech recognition.

Unlike traditional ML models SVM and ANN, which have only a shallow infrastructure to encapsulate features, DL models have used a multi-layer structure to uncover interesting patterns and nonlinearities. Layers capture features from various angles before establishing a multi-level abstraction [14,15]. One can anticipate the promise, widespread

use, and influence of DL integration with extensive mobility data in the predictions of future traffic. This research explores an overview of the fundamental elements that go into the process of predicting road traffic, such as the forms of inputs, mobility data, traffic modeling, and numerous target traffic indications, including speed, flow, and traffic conditions [16–18]. While discussing early initiatives that have already tapped into DL for accurate forecasts of different traffic indicators, this research explores the potential methods for applying DL to several types of traffic prediction [19,20].

The network traffic monitoring is implemented using a Ryu controller in a software-defined networking (SDN) environment. The study focuses on improving the efficiency and accuracy of network traffic monitoring, a critical aspect of network management [21–24]. The proposed approach utilizes a Ryu controller to collect and analyze network traffic data and provide real-time insights into network performance. Overall, this research provides valuable insights into the potential of SDN and the Ryu controller in improving network traffic monitoring [25–28]. The research demonstrates how the NN can be implemented on a DSP to achieve real-time processing of traffic sign images. The study shows how the model can predict traffic flow and identify areas where congestion is likely. The results demonstrate that the model can help reduce traffic jams by optimizing traffic flow and identifying areas where traffic control measures are needed.

The specific contributions of this research are summarized as follows:

- First, an edge-based vehicular environment is considered to predict road traffic. Edge servers store past historical and real-time information about the user's social media, weather information, road traffic information, and road conditions information.
- Second, multiple features are extracted using DL architecture, i.e., BRNN with the soft GRU, with that information classified into two classes, congested or not.
- Third, an optimization approach is proposed for optimizing the hyperparameters of DL architecture according to the real-time and past traffic data.

The research article is organized as follows. The ideas used in traffic forecasting are first introduced. After that, the reviews of DL in traffic prediction and previous efforts are analyzed, and the potential paths for enhancing the precision and effectiveness of broad traffic forecasting are reviewed.

The experimental findings demonstrate that the suggested model performs better than reference models regarding prediction accuracy and resilience to noise and missing data. Moreover, the sensitivity study checks how different hyperparameters affect how well the model works. Therefore, creating more effective traffic management systems in smart cities with the suggested GRU-based Soft-RNN model can improve people's quality of life and lessen the economic losses of congestion.

The cloud-based vehicular environment system can gather data from various sources, such as GPS devices, traffic sensors, and mobile phones, to comprehensively understand traffic conditions. The data are then processed and analyzed using ML algorithms and predictive models to identify patterns and predict congestion in Figure 1.

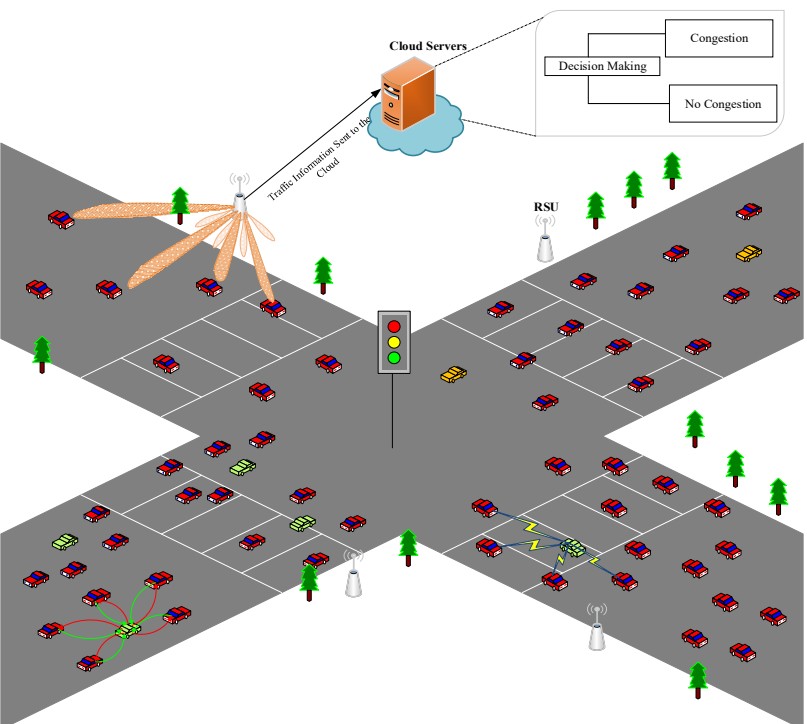

**Figure 1.** Cloud-based Vehicular Environment (Traffic Congestion Prediction).

## 2. Preliminary Knowledge

Some of the established techniques used for motion prediction are described below.

Statistical Approach: Statistical methods can detect traffic patterns at different scales, e.g., during the day, on different days of the week, across the seasons, etc. They are generally faster, cheaper, and easier to manufacture than ML. However, they are less reliable because they cannot analyze as much multivariate data. Since the 1970s, autoregressive integrated moving average models (ARIMA) have been easier to use to anticipate traffic jams and have been more precise than other statistical methods. To examine the past and forecast the future, traditional statistical approaches have been applied. They collect data at regular intervals and assume that current trends will continue. However, one-dimensional ARIMA models cannot handle complex structures and the many variables that alter traffic flow.

ML approach: ML allows the creation of predictive models considering large amounts of heterogeneous data from multiple sources. Research has been conducted on the use of ML algorithms for traffic estimation. The random forest (RF) method creates many decision trees and integrates their data to develop accurate predictions. With sufficient training data, effective results can be achieved quickly. This approach has demonstrated 87.5% accuracy when used to resolve congestion. The model's input parameters include weather, time, specific traffic conditions, road infrastructure, and holidays. To predict possible trends, the k-nearest neighbor (KNN) method uses the idea of the similarity of features. Studies using the ANN model have shown that it can predict short-term traffic flow with an accuracy of over 90%.

DL approach: When comparing DL techniques to ML or statistical models, DL methods consistently show at least 90% prediction efficiency. NNs are the basis of DL algorithms. Artificial NNs (ANNs) use interconnected nodes, or neurons, arranged in two or more layers and are designed to mimic the behavior of the human nervous system. Many types of NNs have been developed for different applications. Convolutional NNs (CNNs) are recognized as industry pioneers in image recognition and analysis. Using road surveillance camera footage, congestion monitoring is a logical application for infrastructure problems. In this case, the average categorization accuracy is 89.5%. CNN would not be the first choice for traffic forecasts.

In contrast, the intention to create a CNN-based model capable of predicting transit speeds has succeeded. To do this, the scientists created a two-dimensional image matrix based on temporal and spatial data that characterize traffic flows. Unlike CNNs, RNNs are designed to analyze data from time series or experiments collected over specific periods. Motion patterns are an excellent example of such observations. When using RNN models, studies have shown high accuracy in predicting congestion development. However, their disadvantage is the dispersion gradient problem, which causes some data from previous layers to be lost. Because of this "forgetfulness", learning the algorithm is more complex and takes longer. Variations of RNNs that address the dispersion gradient problem include long-term memory (LSTM) and GRU. Studies comparing the performance of these models have shown that the GRU model is easier to train and more accurate at predicting traffic levels. Many studies recommend developing various NNs models for motion prediction, including graph NNs, fuzzy NNs, Bayesian NNs, and others, and hybrid techniques that integrate two or even more algorithms. No perfect technique can be used in all situations to obtain the most accurate projections. So, first, let us look at what information is needed for traffic prediction and where to find it. For accurate predictions, all variables affecting traffic must be considered. Therefore, many important categories of data must be collected to anticipate the movement and display of data. First, a complete road network map is needed with the associated attributes. Connecting to global geographic data sources such as Google Maps, TomTom, HERE, or OSM is a great idea to get a clear picture of the current data.

INRIX Dataset: The INRIX dataset contains real-time and historical traffic data collected from GPS-enabled devices such as smartphones and navigation systems. It covers more than 200 countries and includes traffic speed, travel time, and congestion information from https://www.inrix.com/solutions/data/ (accessed on 19 October 2022). The other two datasets were also utilized in this research from https://pems.dot.ca.gov/ (accessed on 19 October 2022) and https://www.inrix.com/solutions/data/ (accessed on 19 October 2022).

*Motivation and Application*

Predictive modeling strategies are used to reduce traffic congestion in smart cities. DL systems may be trained on previous traffic data to identify trends and forecast future traffic conditions. This method can assist city planners and transportation officials to predict congestion before it happens and take preventative measures to avoid it. For instance, they may alter the timing of traffic lights, reroute traffic to less crowded locations, or warn motorists to stay away from specific places at particular times.

## 3. Literature Review

The construction of Intelligent Transportation Systems (ITS) is currently widespread worldwide. As a critical component of ITS, traffic congestion forecasting gives travelers reliable traffic data to save time and helps transportation management organizations handle the road network. Due to the complicated connection between road segments, researchers noted that the current studies, particularly, NNs, do not function well. Additionally, the impact of traffic congestion prediction is worsened by the absence of a higher evaluation criterion for congestion. Researchers provide a technique in this work for mining free-stream speeds and free-stream flow to produce traffic congestion scores. Researchers suggest a road network grouping strategy based on association subgraphs to pre-train DL models and realize data exchange among road segments while considering the road segments' association properties in the road transport system. A traffic congestion prediction model called SG-CNN combines the characteristics of traffic conditions and the CNN model. The process of training is optimized by the road network grouping method. The results of the trials demonstrate that it is superior to other methods in terms of accuracy. The city's ongoing development has made the traffic issue more and more critical. This model suggests a straightforward traffic congestion forecast approach based on the RF algorithm to lessen the discomfort of people's travel resulting from traffic congestion. The degree of traffic

congestion is first determined using DBSCAN. The long-term average speed and traffic flow of urban roadways are trained and predicted using the RF algorithm. The combined model is used to anticipate the level of traffic congestion. The testing findings reveal that the precision is 94.36%, which shows the method's effectiveness. The experiment used traffic data from high-speed roads in the PEMs dataset of the United States.

These traffic jams not only result in a significant number of fatalities and high pollution levels, but they also slow economic growth by restricting the movement of people and commodities, raising the loss of working hours, and raising fuel usage. Numerous research projects have successfully concentrated on foreseeing traffic congestion and then foreseeing their patterns to address this issue. Despite their applicability, the suggested remedies to traffic jam spread have relied heavily on past data. Additionally, they have not effectively allocated traffic control resources based on their estimates. A two-stage traffic resource dispatching approach is developed to create a self-organizing traffic control system enabled by the IoT. This method models and forecasts the distribution of traffic jams throughout a road network using a Markov Random Field (MRF) at its first stage. The method employs the Markov Decision Process (MDP) to autonomously distribute the assets for road traffic following the predictions obtained.

The rapid development of AI applications offers unmatched prospects to raise the effectiveness of many systems. For example, the transportation industry faces additional challenges due to adopting and fusing various global vehicles and environmental issues. Due to the sharp increase in vehicles on the road, road traffic is one of the most pressing challenges in this respect. This model suggests a cloud-based intelligent road traffic congestion prediction framework equipped with a hybrid neuro-fuzzy technique to solve this enormous problem. The study aims to shorten the wait times drivers encounter at various city traffic intersections.

In a smart city environment, observation-based data are collected from multiple embedded IoT sensors along the street. This approach also aimed to support autonomous traffic control systems by reducing congestion. It uses the data by adding a neuro-fuzzy algorithm after appropriate preprocessing by a cloud server. As a result, it has a high level of accuracy through intelligent decision making with a low error rate. After experimental simulations, the accuracy of the proposed model was 98% during the validation test which is higher than the highest accuracy rates reported in the literature for prior art methods, which were 90.6%, 95.84%, 97.56%, and 98.03%, respectively. The trajectories were used to capture better and map Nepal's road infrastructure. Task scheduling has been used in parallel computing to speed up calculations and better use computing resources [29]. This study uses RF to build a traffic prediction model. The high stability, excellent reliability, and high accessibility define the RF algorithm. Input factors such as weather, time of day, season, abnormal road conditions, traffic conditions, and holidays are used to create a traffic prediction model. The results show that the traffic prediction model developed with the RF classification method has an accuracy rate of 87.5%, the generalization error is small, and it can be predicted successfully. Moreover, the calculation speed is fast, which is more helpful in predicting overload conditions. The spatiotemporal context integration with a metric learning approach (STE-ML) is developed to predict traffic congestion intensity. STE-M comprises a feature learning component and a traffic spatial–temporal context embedding element. The context embedding component can integrate regional spatial–temporal correlation characteristics and worldwide traffic statistics data concurrently, compressing them into a single, abstract embedded model. This is possible from both local and global viewpoints. While this happens, the metric learning aspect gains from mastering a more suited distance function tailored to a particular activity. These models are combined to improve the accuracy of traffic congestion predictions [30]. For an ITS in a smart city, an accurate traffic flow forecast is crucial. Accurate traffic congestion forecasting helps with sensible urban management and strategic energy use concerning the problem of traffic congestion. Data-driven traffic flow congestion prediction has various drawbacks, such as erroneous predictions from complicated spatiotemporal correlation patterns. To solve

this issue, an attention-based spatiotemporal attention combination network (STACN) is developed for predicting traffic congestion. First, the multi-dimensional time series relationship of the target road is captured using the standard attention technique. Second, the spatial dependency of each neighborhood along the target route is captured using the graph attention technique. The actual traffic dataset is used in this study to examine the forecast accuracy and consistency of the congestion level, and the experimental findings support the model's efficacy.

The effectiveness and capability of a road network can be increased, and traffic can be avoided by accurately forecasting the amount of congestion. Despite its importance, academics and traffic engineers do not like to anticipate traffic congestion. There are not enough effective computer techniques for traffic flow prediction or elevated citywide traffic data. The Seoul Transportation Operation and Information Service (TOPIS), an accessible online web system, and a hybrid NN LSTM, transposed CNN, and CNN is combined to retrieve the spatial and temporal information from the source image, respectively. The road congestion intensity is forecasted using processed GPS trajectory data because speed sensors are still not used as often as GPS trackers. The average speed of road segments can be calculated using nearby GPS trajectory data, and a hidden Markov model is utilized to link GPS data collected to the road system.

The congestion level forecast uses four DL models—CNN, RNN, LSTM, and GRU—and three traditional ML models: autoregressive integrated time series, support vectors, and peak regression. Experimental results show that DL models outperform traditional ML models regarding traffic volume prediction accuracy. ITS offers real-time traffic services to increase client convenience. Using these services helps spread out traffic and ease congestion. Next, it sacrifices accuracy for the convenience of these services. Since these services often rely on measurement data, data collection will determine the accuracy of the models. As a result, the LSTM prediction approach corrects for missing spatial and temporal variables. The prediction approach first performs preprocessing, including removing outliers using average standard deviations of traffic data and fitting temporal and geographic values using temporal and spatial data patterns and trends. Data with time series aspects have not been adequately trained in previous studies. The prediction technique trains the LSTM model on the time series data. The success of this method was evaluated by determining the mean absolute error (MAE) and comparing it to other models. With a MAE of around 5%, this method was the best of the models compared. Congestion and potential delays at intersections can be significantly reduced through effective traffic management and traffic light placement. Accurately anticipating traffic ahead of the next cycle is an essential aspect of traffic light timing optimization. A perfect approach cannot be achieved with an inaccurate prediction of incoming traffic. This method uses DL to create a real-time, data-driven queue length prediction technique. Imagine a network corridor where sensor data are sent from vehicles (located at an intersection) to subsequent intersections. According to the hypothesis, the duration of the delay at the intersection of the next bike lane is influenced by the duration of the congestion at the destination intersection and the two upcoming intersections.

## 4. Research Gaps

For a few decades, road traffic prediction has drawn increased attention. Each country has been experiencing traffic congestion challenges due to the building of infrastructure. As a result, forecasting the congestion can help the authorities prepare and take the appropriate steps to avoid it. Researchers have started using several models in this area as a result of the advancement of AI and the accessibility of extensive data. Probabilistic models are generally straightforward, but when many elements, such as the weather, social media, and events, which influence traffic congestion, are considered, they become more complicated. In this situation, ML, profound learning, is advantageous. Because they can evaluate an extensive dataset, DL algorithms have grown in popularity over time. The amount of research on predicting traffic congestion is growing tremendously. Most investigations

employed static sensor or camera data from the two sources. Whereas sensor data cannot capture dynamic traffic changes, numerous source changes make assessing flow patterns for probe data challenging. An essential consideration in studies of traffic congestion is the data-gathering horizon. As traffic is dynamic, the short time frame of a few days cannot depict the actual state of the bottleneck. The seasonality's limitations were demonstrated by other research that employed data spanning a few months [31–35].

The environment significantly influences traffic congestion. Several research studies have concentrated on these elements. Five studies considered weather, while two studies considered social media contributions. National holidays, popular sporting events, and public holidays contribute significantly to traffic congestion. As an illustration, Melbourne, Australia, observes two national holidays before and after the nation's two most important sporting events. To deal with the traffic and the parade, the officials restrict a few traffic routes, which causes congestion. Hence, it is essential to pay more attention to integrating these elements in forecasts [36–40].

## 5. Proposed Methodology

The increase in vehicle routine creates the need for effective traffic management as the traffic pattern differs in each area. This technology enables drivers to share more information, including weather conditions, traffic information, directions, accident information, information about the buildings such as hospitals, etc. and many others directly without phone calls. With the increasing population development in urban and rural regions, the necessity for appropriate transportation networks that can provide good compatibility for road users has become a key concern in India. Level of Service (LOS) is one such compatibility metric that provides a quality measure for the operating circumstances within a traffic stream, i.e., the service the road provides to the user. This research looks at the many levels of service models for urban and rural roads that scholars worldwide have proposed. There are several generally used ways to determine the level of service of a given road stretch, such as cluster fuzzy set theory, genetic algorithms, and analysis. This research discusses and reports on techniques such as NNs. A new method for determining the degree of service was established, namely, utilizing the volume-to-capacity ratio, average vehicle speed, and percentage speed reduction.

This research synthesizes LOS data and conclusions using analytical models that may estimate levels of compatibility among diverse road users in an urban environment with various traffic conditions. This research aims to investigate LOS for various routes that have been identified and debated. Bhimavaram (India) town, Ramayanpuram, and Jaganadhapuram villages were used for research. Both study areas-1,3 and 2,4 received a LOS-D rating, while study area-2,4 received a LOS-E grade. Some corrective actions were implemented to improve the LOS grades and traffic conditions: road traffic trajectory and congestion in rural and urban areas using DL RNN.

The architecture of the road traffic system comprises several key components that work together to ensure the smooth and safe movement of vehicles and pedestrians on roads in Figure 2. BRNN combines the two hidden layers of opposite directions to a similar output; hence, the output layer includes backward and forward states, concurrently. The traditional RNN does not predict future information from the current state. BRNN overcomes this. The forward layer is used for the positive direction, and the backward layer is used for the negative layer. The forward and backward hidden layer of the BRNN is defined as follows.

$$H_f = \delta(x(t)w(f) + H(t-1)w(f) + b(f)) \tag{1}$$

$$H_b = \delta(x(t)w(f) + H(t-1)w(b) + b(b)) \tag{2}$$

where $x(t)$ represents the input, and $w(f)$ and $w(b)$ represent the weight values of the forward and backward hidden layers, respectively. $b(f)$ and $b(b)$ represent the bias parameters. Then, the BRNN concatenates the two hidden states, forward and backward, for final outputs. The final output of the BRNN is defined as follows.

$$O(t) = H(t)W(c) + b(c) \tag{3}$$

where $W(c)$ represents the weight matric of the final output, and $b(c)$ represents the bias parameter of the output layer. The training process of BRNN is defined as follows.

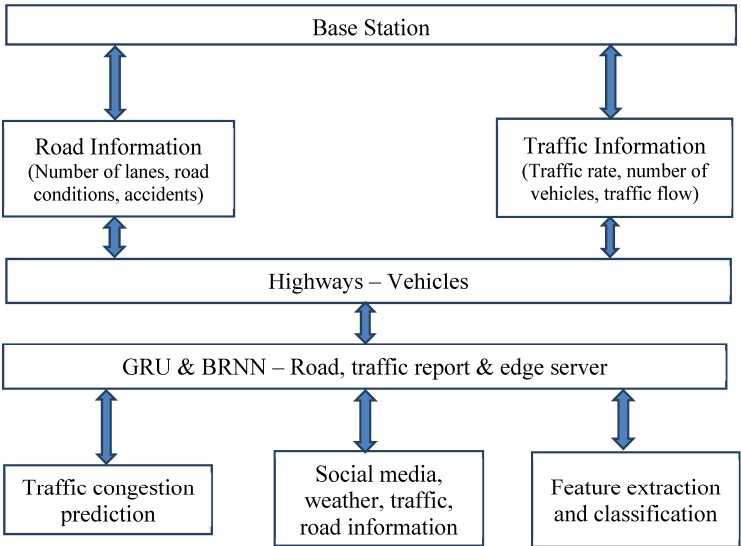

**Figure 2.** System Architecture.

Algorithm 1, Bidirectional LSTMs have the ability to comprehend prolonged dependencies within a sequence, rendering them appropriate for tasks that demand an understanding of context over extended durations.

---

**Algorithm 1.** Bidirectional LSTM

---

Load the data
Initialize the batch size, number of steps, and number of devices
Training the iterations with load data time
Define the bidirectional LSTM model for setting bidirectional is equal to true
Define the number of hidden layers, and the number of layers
Process LSTM layer (LSTM.LSTM (No. Hidden, No. layers, Bidirectional=true)
Update the result of the RNN model
Train the procedure from steps 1 to 7
Define the number of iterations
Provide training results of BRN

---

### 5.1. GRU

This section explains the general process of GRU. It is the advanced version of Standard RNN. The LSTM includes three gates that do not maintain the internal cell state but are integrated into the hidden state of the gated RNN. This information is transferred to the next GRU. The various gates of GRU are defined as follows.

Update gate: It defines how much previous knowledge needs to be forwarded to the future. It determines the corresponding Output Gate in an LSTM recurrent unit. It is calculated using

$$y = \sigma\left(W^{(y)}Z_t + V^{(y)}H_{t-1}\right) \tag{4}$$

where $Z_t$ is given to the network unit, which is multiplied by the weight value $W^{(y)}$. That is forwarded to the hidden layer $H_{t-1}$, which includes the information of the previous states, and is multiplied by its weight values $V^{(y)}$. These two results are added to provide the final result in the update gate between 0 and 1. It represents how much past information wants to be forwarded to the future. It is used to eliminate the risk of gradient problems.

Reset gate: It represents how much of previous knowledge can be forgotten. It corresponds to the input gate and forgets it in an LSTM recurrent unit. The reset fate calculation is defined using

$$R = \sigma\left(W^{(y)}Z_t + V^{(y)}H_{t-1}\right) \tag{5}$$

The two results are added and multiplied with their weight values, and the sigmoid function value is applied to the output results.

Current memory gate: This gate is integrated into the reset gate such as the input modulation gate, a subpart of the input gate. It provides the nonlinearity input and also provides zero mean input. It is used to reduce the outcome that previous information of the current information is forwarded to the future gate. The calculation of the current memory gate is performed using

$$H = \tanh(Wz_t + R \odot VH_{t-1}) \tag{6}$$

Multiply the input $z_t$ with its weight value, and the hidden state is multiplied by its weight value. Then, compute the Hadamard product between the reset gates $R \odot VH_{t-1}$. Then, add the output of the first and second process values. For this output, tanh nonlinear activation function is applied to calculate the current memory content of the GRU. Finally, calculate the hidden state value that includes the current unit's values and then forward it to the network, performed by the update gate. It determines the current memory content. The calculation of current memory is defined using

$$H_t = y_t \odot H_{t-1} + (1 - y_t) \odot H_t \tag{7}$$

Perform element-wise multiplication to the update gate and then calculate element-wise multiplication to $(1 - y_t) \odot H_t$. Finally, add these two results for calculating the current memory content of the GRU. Figure 3 represents the general architecture of GRU. GRU enhances the RNN memory capacity and solves the vanishing gradient problems. It is suitable for many applications such as machine translation, speech signal modeling, recognizing handwriting, etc.

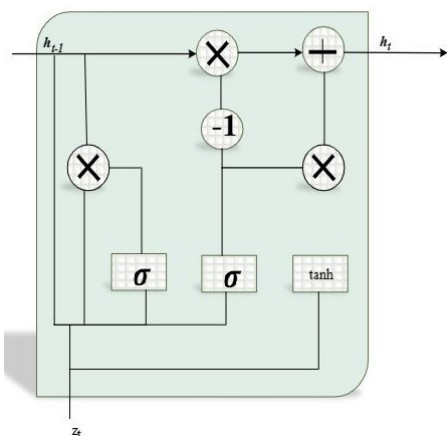

**Figure 3.** General Architecture of GRU.

The key innovation of GRUs is the use of gating mechanisms to control the flow of information in and out of the cell. The gating mechanism consists of an update gate and a reset gate. The update gate controls how much of the previous state should be retained, while the reset gate controls how much new input should be incorporated into the state in Figure 3. The proposed bidirectional RNN includes two layers of RNN that can process concurrently. Y is input and $Y_k$ denotes the different inputs with different timestamps. The processing is performed concurrently, and the layers are arranged successively. The input is processed one by one in the other layer. The hidden layer of RNN includes two hidden states determined for every time step. The hidden layers are combined into one

layer by adding two inputs using a simple addition operator. Bidirectional RNN executes each neuron of the network. The feature-matching process considers the input and output elements for processing the time step. RNN includes Soft-GRU, has low complexity, and processes the input $\widetilde{x}_{k,n}$ using history $h_{k-1,n}$. This process has low complexity due to the activation function and soft plus. The proposed Soft GRU is defined as follows.

$$h_{k,n} = (1 - z_t) \odot h_{k-1,n} + z_t \odot \widetilde{x}_{k,n} \tag{8}$$

$$\widetilde{x}_{k,n} = \pi(W_x x_{k,n} + b_x) \tag{9}$$

$$z_t = \sigma(W_x x_{k,n} + U_z x_{k-1,n} + b_z) \tag{10}$$

where $\pi(x)$ represents the function of soft plus, which is computed as $(1 + e^x)$ and $\sigma$ represents the sigmoid function and $b_x$, $b_z$ represent the values of biases, and $W_x$, $W_z$, and $U_z$ represent the weights values of the GRU. The process of feature extraction using bidirectional RNN is depicted in Figure 4. GRU can operate for natural language processing, which utilizes past and real-time information about social media, traffic, road, and weather, presented in the edge servers. Figure 5 represents the flowchart for the overall process of traffic density prediction.

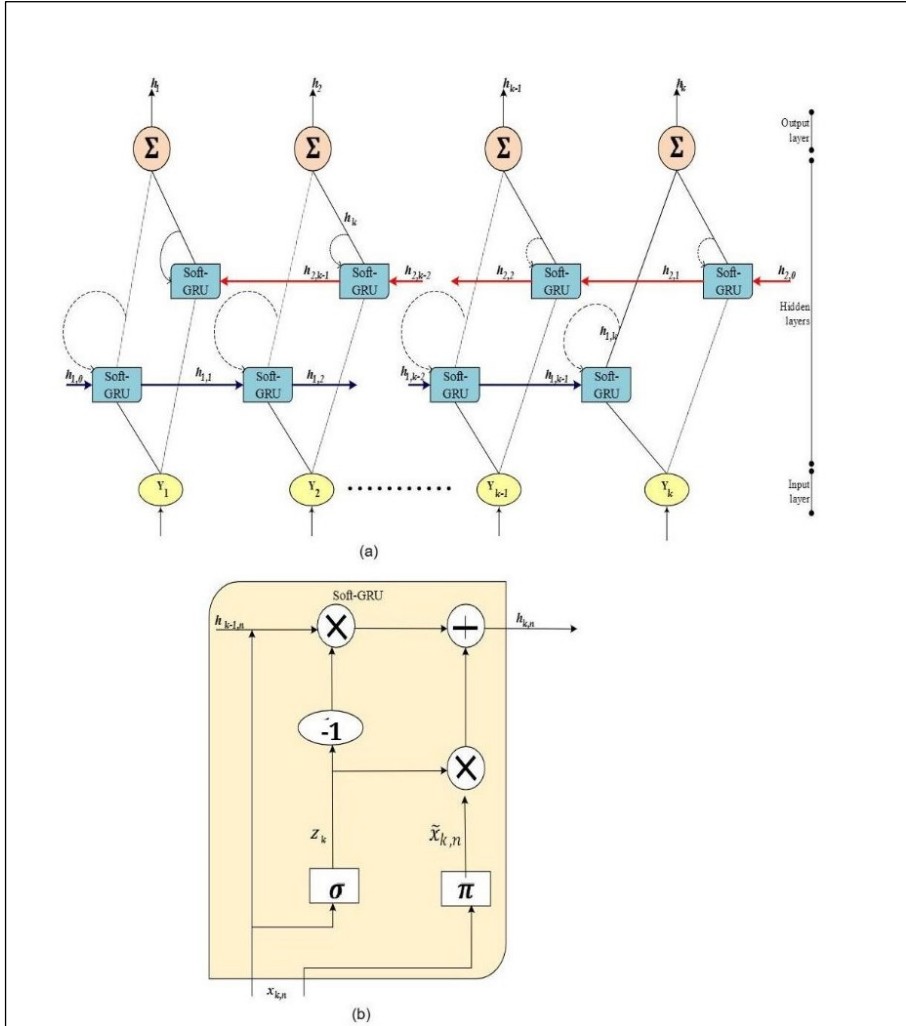

**Figure 4.** (**a**) Bidirectional RNN with layers. (**b**) Architecture of bidirectional RNN using soft GRU.

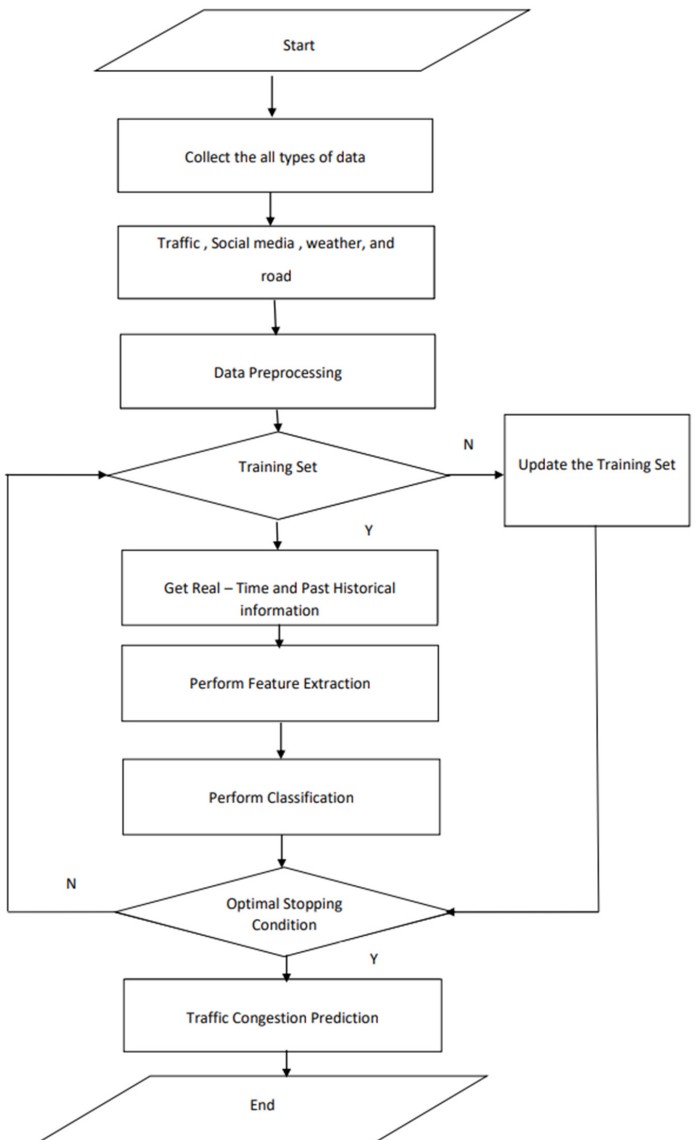

**Figure 5.** The overall process of Traffic Congestion Prediction.

### 5.2. Traffic Information

Next, the data are collected on past and present traffic conditions, including the amount, sort, and velocity of cars that pass through a specific location (trucks, light vehicles, etc.). These data are gathered using camera systems, loop sensors, and other devices.

### 5.3. Data on the Weather

Historical, present-day, and anticipated weather data are required because weather factors affect traffic and speed limits on the road.

### 5.4. Traffic Density

The number of vehicles on the road about its length is called traffic density. Due to the lack of sensors to gauge the existence of vehicles, it has historically been challenging to determine the traffic volume for the entire road length. However, this tendency is shifting with more traffic cameras and advancements in ML. Occupancy is a related term frequently used as a stand-in for density measurement. The proportion of time that a particular location on the road network has vehicles parked there is known as occupancy. Sensors can measure usage, making vehicle loop detectors the most popular choice (VLDs). In a homogenous stream of traffic, each vehicle has the same length, occupancy, and traffic

flow, and they are precisely related. The fundamental relationship between density and speed (q = ku), where q is the flow, k is the density, and u is the speed, is most frequently used in practice to determine density. The connection between occupancy versus density is complicated when there is a heterogeneous traffic stream. In a BRNN using a soft GRU, the input sequence is first fed through a forward GRU layer and then through a backward GRU layer. The outputs from both layers are then concatenated to create the final output sequence in Figure 4.

Traffic Velocity (speed): The pace recorded at a particular time and place is known as a vehicle's spot speed (or instantaneous speed). This is the pace that the car's speedometer registers. In engineering management, however, our focus is on figuring out mean speed because it can be utilized to define the movement of vehicles. The aggregate can be used in either time or space to calculate mean speed.

The space-mean speed for a particular space interval is calculated as the ratio of the total distance traveled by all vehicles to the total time needed. The average of each of the two vehicles' specific speeds is the time-mean speed for a specific period. The process of traffic congestion prediction involves collecting data, preprocessing the data, engineering features, selecting and training a suitable model, evaluating the model, and deploying it to predict traffic conditions in real time. This process can help improve traffic management and reduce roadway congestion in Figure 5.

### 5.5. Traffic Flow

Traffic flow is the number of vehicles crossing a specified reference location at a given moment. Usually, a section's center or end is used as the point of reference.

### 5.6. Experimental Results

In this section, an edge-assisted vehicular environment is evaluated regarding performance metrics. The performance achieved by the proposed method is compared with previous research works. The proposed edge-based vehicular environment is modeled using an integrated simulation system incorporating the OMNeT++ and SUMO traffic models. INETMANET is used in this case to design the ad hoc network.

An open-source model library called INETMANET was created for the OMNeT++ simulation environment. Both necessary wireless protocols and ad hoc network protocols are supported. The comprehensive simulation environment is built to deliver a realistic vehicle movement trace that can be utilized in the real world. The network is simulated using OMNeT-4.6, SUMO-0.21.0, and INETMANET-2.0. INETMANET is an open-source network simulation framework that can simulate various network systems, including road traffic networks. It can benefit road traffic networks in multiple ways, such as evaluating performance, managing traffic, analyzing safety, and assessing environmental impact.

Figure 6 illustrates the general simulation model of an edge servers-assisted vehicular environment. Here, shaded areas denote obstructions, and black lines signify traffic lanes. The shapes in the yellow color depict vehicles. Firstly, we compile the geographic information regarding the location of traffic monitoring stations along the highway's sections, the road system geometry file, and the time data in the form of traffic flow captured at each sampling interval (3 min time intervals in our case). SDN, Cognitive Radio (CR), and Vehicular Ad Hoc Networks (VANETs) technologies (SDN-CR-VANET). SDN-CR-VANET technology aims to enhance the efficiency and dependability of VANETs by utilizing radio spectra more effectively and managing the network centrally. The technology allows for resource allocation based on traffic demand, while the CR technology dynamically adjusts transmission parameters to optimize spectrum utilization. In addition, SDN-CR-VANET technology has a broad range of applications, including traffic management, emergency response, and entertainment.

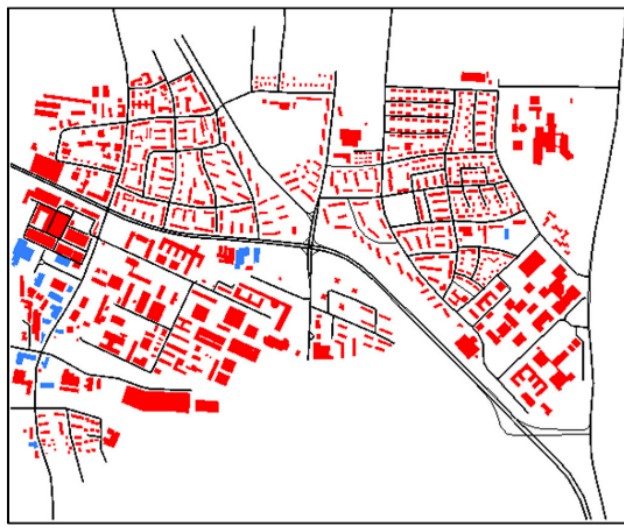

**Figure 6.** Simulation environment of SDN-CR-VANET.

In Table 1, several approaches have increasingly highlighted the significance of hyperparameters for the training phase. This research considers five hyperparameters—learning rate, number of RNN layers, number of nodes, batch size, and dropout—for tuning in RNN models concerning the traffic data from the highway system. This one is predicated on our prior works. In particular, Table 2 shows the search spaces and descriptions for the abovementioned hyperparameters. The focus of this research is to reduce the square error for estimating traffic flow by using the automatic hyper-parameters tune challenge, and this goal is stated as follows:

$$E = \sum_{i=1}^{n}(Y_i - P_i)^2 \tag{11}$$

E represents the square error values, and P represents traffic flow values. Similarly, other certain metrics, including Mean Absolute Error (MAE), Mean.

where

$Y_i$ represents the actual value or target value for observation i.

$P_i$ represents the predicted value for observation i.

$(Y_i - P_i)^2$ is the square of the difference between the actual and predicted values for observation i.

$\sum$ the symbol means summating all the squared differences across all n observations from i = 1 to n.

**Table 1.** Comparison Results for Proposed and Existing Methods.

| Unique Aspect | Data Source | Performance | DNN Architecture | Congestion Is Defined on the Basis of: | Paper |
|---|---|---|---|---|---|
| Efficient encoding for spatial information | 11 intersections (VLDs) 3 months Florida, USA | RMSE~1 | LSTM | Queue length | (Rahman and Hasan, 2020) [32] |
| Scalable architecture | Speed heat map Seoul, S Korea | Accuracy: 84.2% | Novel PredNet) (built using CNN&LSTM) | Traffic speed | (Ranjan et al., 2020) [29] |
| Congestion tree | 553 road links (5 weeks) Helsinki, Finland | MSE: 0.73 (weekdays), 0.37 (weekend) | Conv-LSTM | Not applicable (pre-labeled by data provider) | (Di et al., 2019) [33] |
| Detailed sensitivity analysis with regard to the input horizon | 2000 taxis GPS (28 days) Chengdu, China | 90.55% ≤ Accuracy ≤ 96.32% 91.89% ≤ Accuracy ≤ 96.75% | CNN LSTM | Traffic speed | (Sun et al., 2019) [30] |
| Observation: The sort of network affects how complicated a task is. | Seoul, South Korea's metropolitan suburbs and the surrounding area | MAPE: 4.29% (urban) MAPE: 6.08% (suburban) | LSTM | Traffic speed | (Shin et al., 2020) [31] |

**Table 2.** Hyperparameters Tuning Bi-Directional RNN with Soft GRU.

| Parameter | Explanation | Data Type and Values |
|---|---|---|
| No of Neurons | The units within the hidden layer's techniques for accuracy maximization | Log Uniform or Int [1, 200] |
| Dropout | Minimizing the overfitting of neural nets | Floating [1, 0] |
| Learning Rate | Error-values are adjusted according to the weight values | Log Uniform/Floating [0.1, 0.2, 0.005] |
| Hidden Layers | Input and output layers that maximize the accuracy | Int [0, 2] |
| Batch Size | Describing the no of samples that propagates via the process | Int [1, 512] |

MSE and Root Mean Squared Error (RMSE) are frequently used to measure efficiency for regression problems.

$$MSE = \frac{1}{N} \sum_{i=1}^{N} \left( A_i - \widehat{PR_i} \right)^2 \tag{12}$$

$$RMSE = \frac{1}{N} \sum_{i=1}^{N} \left( \widehat{A_i} - PR_i \right)^2 \tag{13}$$

$$MPSE = \frac{1}{N} \sum_{i=1}^{N} \frac{\left( A_i - \widehat{PR_i} \right)}{A_i} \tag{14}$$

where $A_i$ and $PR_i$ are the actual and predicted values, respectively.

MSE: Mean Squared Error, a standard metric used to measure the average squared difference between a dataset's predicted and actual values.

RMSE: Root Mean Squared Error.

N: the total number of data points in the dataset.

$A_i$ : the actual value of the i-th data point in the dataset.

$PR_i$: a model or algorithm estimates the predicted value of the i-th data point in the dataset.

i: the index of the data point in the dataset.

DL is a complex problem because of the numerous hyperparameters they contain. In particular, changing every hyperparameter can complicate the procedure. Additionally, search algorithms consider hyperparameters equally to discover the better effect, which might cause time-consuming and complex issues to compute. The formulae show that RMSE and MAE are based on the unit, whereas MAPE is a dimensionless variable. Whenever MAPE was available, we sought to present it in this survey with the results of several regression exercises. Residency and computational time are important factors to consider in various fields. Balancing these factors can lead to optimal performance and efficiency in a given task or process in Figure 7. As seen in Table 3, a confusion matrix can be utilized to describe the most frequently used parameters for classification. Several metrics are described in the context of the confusion matrix. The three metrics, true positive rate (TPR), true negative rate (TNR), and precision, are provided.

$$TNR = FP + TN \ / \ TN \tag{15}$$

$$TPR = FN + TP \ / \ TP \tag{16}$$

$$Accuracy = \frac{(TP + TN)}{(TP + FN + FP + TN)} \tag{17}$$

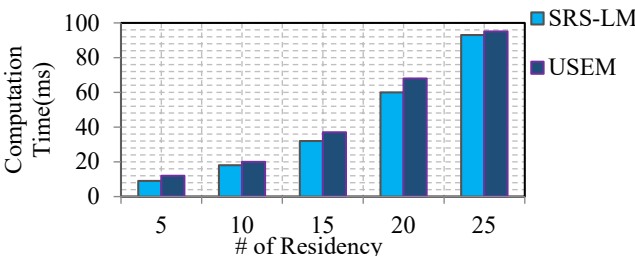

**Figure 7.** Residency versus Computational time (ms).

**Table 3.** Binary classification-based confusion matrix.

| Time Class | |
|---|---|
| **Congested** | **Non-Congested** |
| True Positive (TP)Congested Predicted as Congested | False Positive (FP) Non-Congested Predicted as Congested |
| False Negative (FN) Congested Predicted as Non- Congested | True Negative (TN) Congested Predicted as Non-Congested |

The task ahead frequently dictates the decision of the metric used to evaluate performance. Let us look at an instance where a DL approach categorizes the traffic status as "congested" or "non-congested" to illustrate the point.

MAPE stands for Mean Absolute Percentage Error, a commonly used metric for evaluating the performance of regression models in ML. It measures the average absolute percentage difference between a target variable's predicted and actual values.

The formula for MAPE is:

$$\text{MAPE} = (1/n) \times \sum (|(\text{actual} - \text{predicted})/\text{actual}|) \times 100\% \tag{18}$$

where n is the number of samples, actual is the actual value of the target variable, and predicted is the predicted value of the target variable.

MSE stands for Mean Squared Error, a common metric used in machine learning to evaluate the performance of a regression model. The MSE measures the average squared difference between the predicted and actual values in the dataset.

$$\text{MSE} = (1/n) \times \Sigma (y - \hat{y})^2 \tag{19}$$

where

n is the number of data points;

y is the actual value;

$\hat{y}$ is the predicted value.

For finding the computational time, we have to consider different factors such as—determining the size of the input data, determining the number of hidden layers and the number of neurons per layer, determining the type of activation function used in BRNN, determining the type of optimization algorithm used in BRNN, and determining the processing power of computer or server

Once these factors are determined, the computational time of BRNN is estimated using the following formula:

$$\text{Computational time} = \text{number of computations} \times \text{time per computation} \tag{20}$$

For all three data sources (velocity, capacity, and flows), where we have withheld specific data portions, numerous results are presented in previous efforts to better illustrate the efficiency of the new traffic congestion forecast. The results, particularly for flow and capacity data streams, demonstrate outstanding current traffic forecasting outcomes compared to all other currently used approaches. As well as many other outcomes such as

MAE, MAPE, and RMSE, which present the mean values RMSE and standard deviation results achieved across all stations and all statistics from incoming streams, when both methods are used, the advantages of the proposed road congestion prediction can be seen overall. Consequently, the remaining study results will concentrate on showcasing the influence of DL on precision, and performance was evaluated utilizing both congested and non-congested data streams.

When we forecast even more into the future, every system's prediction efficiency drops as is to be predicted. The Conv-LSTM is the model that performs the worst, mainly because it ignores the temporal and spatial correlations between both the counting stations. The proposed model surpasses all other algorithms in every circumstance, while the hybrid approach RNN with a soft GRU comes in second place. Advanced DL models are the models that perform the best overall. We furthermore offer the graphics for the other two performance indicators that were compared to all models. The variation for all models, as calculated using the MAE and SMAPE, are shown in Figures 8–12. The results of the RMSE investigation that was previously reported yield the same findings. Computation time is determined for predicting traffic congestion to the traffic density.

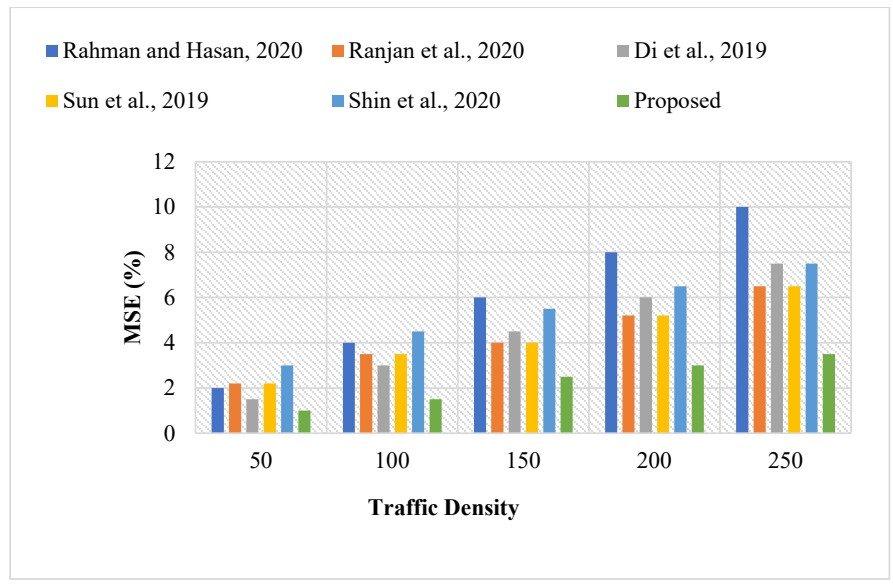

**Figure 8.** Comparative Results for MSE [29–33].

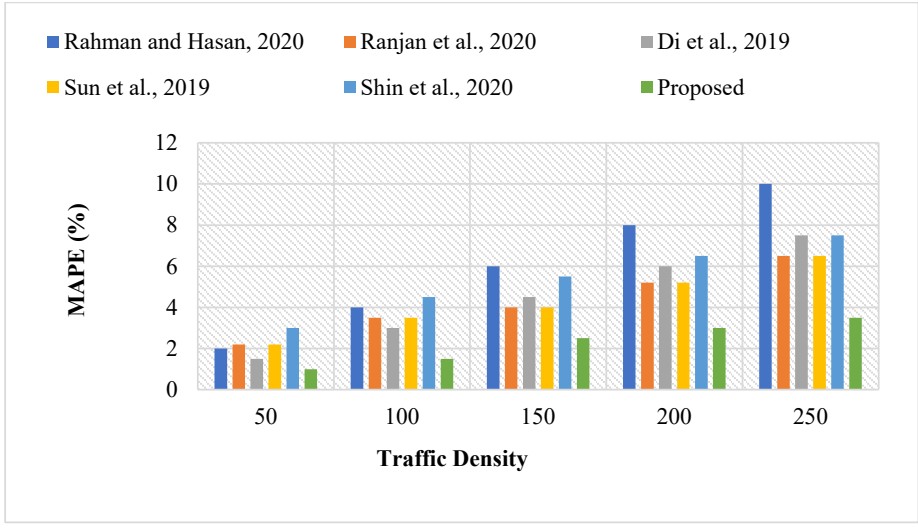

**Figure 9.** Comparative Results for MAPE [29–33].

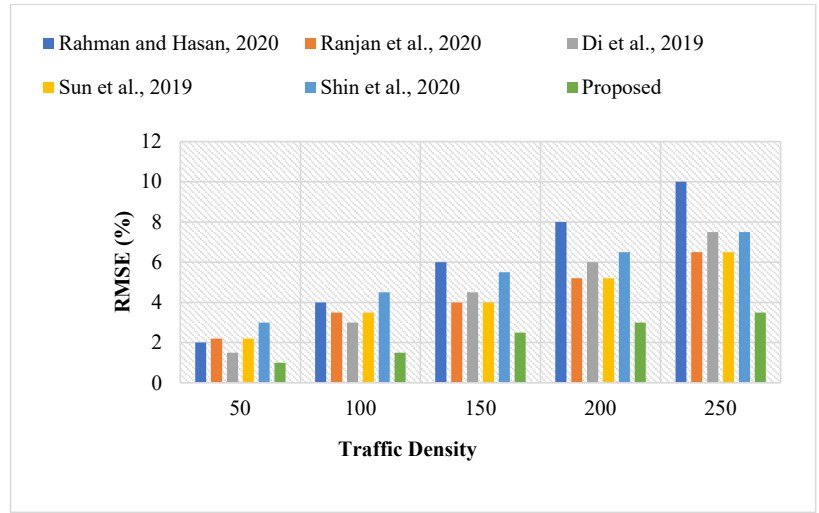

**Figure 10.** Comparative Results for RMSE [29–33].

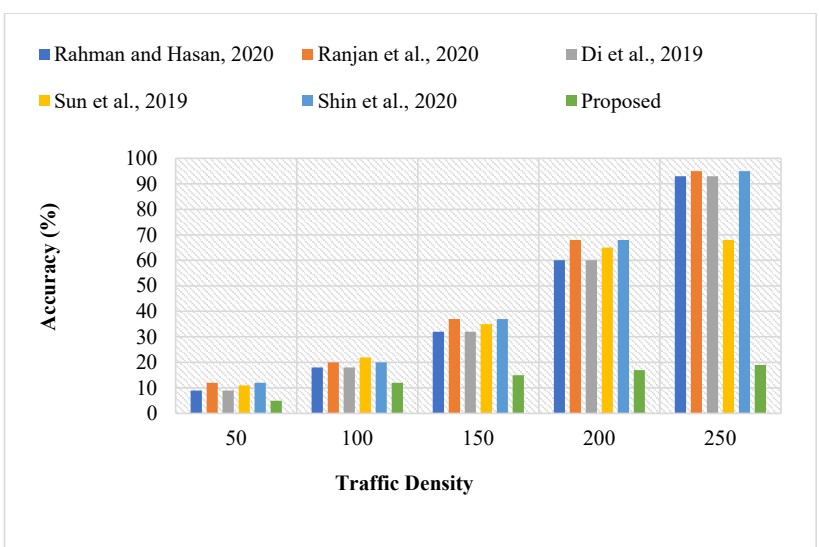

**Figure 11.** Comparison Results for Accuracy [29–33].

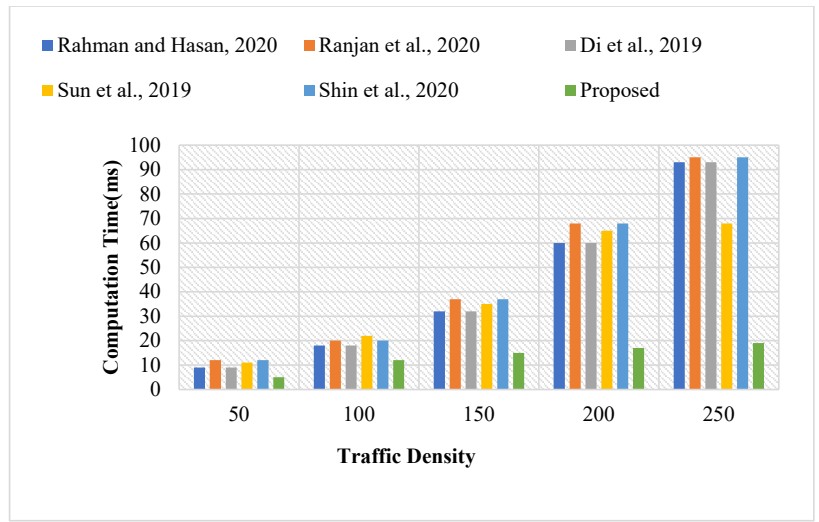

**Figure 12.** Comparative Results for Computational Time [29–33].

Computational time is plotted concerning several increasing traffic densities, as depicted in Figure 12. Computation time gradually increases with the growth in the number of vehicles, since additional involvement of vehicles includes congestion on the road. This comparison shows the computation time for the proposed method decreases while other existing methods have a harder time in predicting traffic congestion.

## 6. Conclusions and Future Work

In this research, we make some speculative claims about the potential use of DL to forecast a range of traffic indicators, including speed, weather, flow, and accident likelihood. The performance of the traffic congestion forecast is enhanced by extracting several features, including the past-day traffic, road, and weather conditions. BRNNs with a soft GRU are suggested to extract traffic data and divide them into two classifications: congested and non-congested. Previous works have been mentioned regarding the cloud environment, which does not guarantee accurate traffic forecasting and which frequently results in accidents on the road. We also summarize and analyze a few recent initiatives with encouraging results. We describe two potential directions for future research to improve the accuracy and efficiency of large-scale traffic predictions. In the future, a semantic matching procedure will be included to predict traffic levels accurately. The proposed framework achieves state-of-the-art performance in detecting COVID-19 and pneumonia, demonstrating the effectiveness of the ensemble approach. The paper contributes to the development of deep learning models for the early detection of COVID-19 and pneumonia, which can aid in the prompt diagnosis and treatment of these diseases. In the time traffic control was maximized that provides safety and health to the human beings [41–43].

The research article proposes a novel approach for traffic prediction in an edge-based vehicular environment. The approach utilizes historical and real-time information from various sources, such as social media, weather information, road traffic information, and road conditions information, stored in edge servers. Multiple features are then extracted using deep learning architecture, specifically bidirectional RNNs with the soft gated recurrent unit (GRU), and classified into two classes, congested or not. An optimization approach is also proposed to optimize the hyperparameters of the DL architecture based on real-time and past traffic data. Finally, this approach has the potential to enhance the precision and effectiveness of traffic forecasting, which can be helpful in various applications such as route planning, traffic management, and congestion mitigation. The article also reviews previous efforts and suggests potential paths for future research in this area. Rahman and Hasan, 2020; Di et al., 2019; Shine et al., 2020; Ranjan et al., 2020; Sun et al., 2019; and our proposed work were compared and depicted in Figures 8–12.

Traffic flow management is a crucial problem in smart cities, and DL methods such as RNNs may significantly enhance congestion prediction. GRU-based soft RNNs have demonstrated promising results in predicting traffic congestion by identifying temporal patterns in traffic flow data. Intelligent city planners may optimize traffic flow by precisely forecasting traffic bottlenecks—modifying traffic lights, or recommending to drivers alternate routes. Decreasing the time that cars are left idle in traffic can also aid in lowering greenhouse gas emissions and improving air quality. To summarize, a possible strategy to improve traffic flow in smart cities is to use GRU-based soft RNNs for congestion prediction. This technology can aid in developing more sustainable and effective transportation systems that are good for both people and the environment.

**Author Contributions:** Conceptualization, S.M.A.; methodology, S.K.R.; software, M.P.; validation, N.A.K.; formal analysis, S.K.T.; investigation, R.M.; resources, A.H.A.; data curation, D.S.K.; writing—original draft preparation, S.K.R.; All authors have read and agreed to the published version of the manuscript.

**Funding:** Princess Nourah bint Abdulrahman University Researchers Supporting Project number (PNURSP2023R120), Princess Nourah bint Abdulrahman University, Riyadh, Saudi Arabia.

**Institutional Review Board Statement:** Not applicable.

**Informed Consent Statement:** Not applicable.

**Data Availability Statement:** In the manuscript URL as given clearly.

**Acknowledgments:** The authors gratefully acknowledge the financial support provided by the Princess Nourah bint Abdulrahman University Researchers Supporting Project number (PNURSP2023R120), Princess Nourah bint Abdulrahman University, Riyadh, Saudi Arabia. We would like to express our sincere appreciation for their generous support and the opportunity to conduct this research. We also thank all the participants who voluntarily participated in this study and made this research possible.

**Conflicts of Interest:** The authors declare no conflict of interest.

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
