# Peer review of "Optimizing Traffic Flow in Smart Cities: Soft GRU-Based Recurrent Neural Networks for Enhanced Congestion Prediction Using Deep Learning"

_sustainability, doi:10.3390/su15075949_

Round 1

Reviewer 1 Report

Author written “Optimizing Traffic Flow in Smart Cities: Soft GRU-based Recurrent Neural Networks for Enhanced Congestion Prediction using Deep Learning”. I would like report few corrections to the author before proceed to the next level.

1. Author mentioned recursive soft gate units (GRUs) in abstract section.  Is it Gated recurrent unit or Gate Recursive Unit? Author need to clarify this.

2. Author mentioned BRNN. What is BRNN ? is it BD-RNN ?. Author can include the final outcome of this proposed model in the abstract and conclusion section.

3. Author presented the soft gated recurrent unit (GRU) but in abstract author pointed soft gated unit . Which one is correct.?

4. In Figure.2 , The system architecture should describe the flow model clearly. Here I could not find the role of GRU and BD-RNN. Its look like general system architecture for traffic control system and congestion control.

5. Figure.4 should be represent as 4a. and 4.b with label name.

6. In Figure.5, conditional statement represents “Training set of Test set” . It’s totally confusing the flow of the entire process. Similarly optimal stopping condition. If its true then its going for update Training set. Later where is the flow ? ie from update the training set where the flow should move ?

7. What is INETMANET ?

8. What is SDN-CR ? Many abbreviations are missing in the manuscript. Author need abbreviate properly in the manuscript.

9. In equation 11,12 and 13 , author need to describe all the parameters usage with clear description.

10. What is TNTP ?

11. Figure.11, Author applied MAPE. What is the statistical formula for MAPE ? What x-axis defining ? its an data or node ? similarly all the figures.

12. What is the computational time formula ?

Reviewer 2 Report

In general, I think it is worthy of publishing. Some points should be included within the manuscript in order to improve the publication.

  • In the introduction section, the current state of the research field should be reviewed carefully and key publications cited and analyzed. A brief description of corresponding studies about image classification would be useful.
  • The data set which is used for training, validating and testing the proposed neural network should be described in more detail.
  • Maybe a figure will help to overview the structure of the proposed neural network.
  • The description of the process of traffic congestion prediction (figure 5) is relatively weak in the present form and should be strengthened with more details and justifications.
  • In the conclusions section, the authors briefly summarize the main points of their study. The authors should explain the contribution of their results in comparison to the results of other researchers. The practical applications of this study should be mentioned.
  • The authors should refer to recent papers, such as the following:

-      Omran M. A. Alssaheli, Z. Zainal Abidin, N. A. Zakaria, Z. Abal Abas, "Implementation of Network Traffic Monitoring using Software Defined Networking Ryu Controller," WSEAS Transactions on Systems and Control, vol. 16, pp. 270-277, 2021.

-     Ninad Patil, Vanita Agarwal, "Performance Simulation of a Traffic Sign Recognition based Neural Network on Cadence’s Tensilica Vision P6 DSP using Xtensa Xplorer IDE," WSEAS Transactions on Computer Research, vol. 10, pp. 35-42, 2022.

-      M. Ali Musri S, Siti Fatimah, Saiful Anwar Matondang, "Simulation Model to Reduce the Traffic Jams with a Stochastic Program," WSEAS Transactions on Environment and Development, vol. 18, pp. 37-41, 2022.

Round 2

Reviewer 1 Report

In Figure.2 , The system architecture should describe the flow model clearly. Here I could not find the role of GRU and BD-RNN. Its look like general system architecture for traffic control system and congestion control

1.      Figure.4 should be represent as 4a. and 4.b with label name.

Author Response

Please see the attachment."

Reviewer 2 Report

The paper can be published in its present form.

Round 3

Reviewer 1 Report

Author answered all the queries. No more comments.